# Concepts in Oncolytic Adenovirus Therapy

**DOI:** 10.3390/ijms221910522

**Published:** 2021-09-29

**Authors:** Klaus Mantwill, Florian Gerhard Klein, Dongbiao Wang, Sruthi Vasantamadhava Hindupur, Maximilian Ehrenfeld, Per Sonne Holm, Roman Nawroth

**Affiliations:** 1Department of Urology, Klinikum Rechts der Isar, Technical University of Munich, 81675 Munich, Germany; klaus.mantwill@tum.de (K.M.); florian.gerhard.klein@tum.de (F.G.K.); dongbiao.wang@tum.de (D.W.); sruthi.hindupur@tum.de (S.V.H.); maximilian.ehrenfeld@tum.de (M.E.); 2Department of Oral and Maxillofacial Surgery, Medical University of Innsbruck, A-6020 Innsbruck, Austria; per-sonne.holm@i-med.ac.at

**Keywords:** adenovirus, cancer, immunotherapy, oncolytic virus, CRAd, vector design, arming, tropism

## Abstract

Oncolytic adenovirus therapy is gaining importance as a novel treatment option for the management of various cancers. Different concepts of modification within the adenovirus vector have been identified that define the mode of action against and the interaction with the tumour. Adenoviral vectors allow for genetic manipulations that restrict tumour specificity and also the expression of specific transgenes in order to support the anti-tumour effect. Additionally, replication of the virus and reinfection of neighbouring tumour cells amplify the therapeutic effect. Another important aspect in oncolytic adenovirus therapy is the virus induced cell death which is a process that activates the immune system against the tumour. This review describes which elements in adenovirus vectors have been identified for modification not only to utilize oncolytic adenovirus vectors into conditionally replicating adenoviruses (CRAds) that allow replication specifically in tumour cells but also to confer specific characteristics to these viruses. These advances in development resulted in clinical trials that are summarized based on the conceptual design.

## 1. Introduction

In recent years, the development of oncolytic viruses and their implementation in clinical trials have gained increased attention. Adenoviruses are among the best-studied viruses in the field of oncolytic virotherapy and several molecular biological discoveries can be traced back to adenovirus research [1,2]. They belong to the class of non-enveloped viruses and contain double-stranded linear DNA with a 26–48 kb genome in an icosahedral capsid of ~950 Å in diameter [3], excluding the fibres, which are between 110 and 370 Å long, depending on the serotype [4]. Human adenoviruses are of high genetic stability and low pathogenicity. They are comparatively easy to produce in high titres and purity. This makes them particularly suitable for various applications, from gene and cancer therapy to vaccine development [5].

The life cycle of adenoviruses is extrachromosomal and they are therefore considered as non-integrating vectors that can infect both replicating and non-replicating cells [6]. According to their antigenic properties, over 50 serologically distinguishable subtypes of human adenoviruses have been described so far [7,8] but recent efforts have revealed 104 genotypes that have been registered (http://hadvwg.gmu.edu/ (accessed on 23 September 2021)). Serotypes 2 and 5 are most commonly used in research.

Gene therapy with replication-deficient adenoviruses is limited to the expression of a transgene and, in the case of suicide gene therapy, to the killing of the infected cell. In the oncolytic approach, on the other hand, the viruses replicate, followed by lysis of the infected cell and release of new infectious units into the immediate tumour environment. An important aspect of adenoviral oncolysis is that it is also effective in tumour stem cells, which often cause tumour recurrences due to intrinsic resistance to therapy [9].

Since the early 1990s, sophisticated methods and advances in molecular biology and virology led to a resumption of virus vector development [10,11]. In particular, the establishment of human embryonic kidney cells (HEK293) that express the adenoviral E1 region was important for the development of first-generation adenoviral vectors that were made replication-defective by deleting the early E1 gene and making room for a transgene sequence of up to 4.5 kb [12]. The E1 region in the HEK293 cell line provides trans-complementation and enables replication of the adenoviral vector, making it to a feasible production cell line.

This review article focuses primarily on strategies of genetic engineering of oncolytic adenovirus vectors for tumour-specific replication, the main prerequisite for oncolysis. It also describes aspects of transgene expression and stimulation of the immune system. These concepts are currently being tested in clinical trials that are summarized based on adenovirus vector design which will determine as a decisive aspect for further improvements in vector design and combination with other treatment options in the future.

## 2. The Four Main Concepts for Conditionally Replicating Adenoviruses Used in Actual Clinical Trials

The replication cycle of adenoviruses can be divided into two phases. Accordingly, the adenoviral genome is organised into early (E1–E4) and late (L1–L5) transcription units, the latter encoding structural proteins (Figure 1). The early region 1A (E1A) encodes five different proteins [13,14], the two main isoforms 12S and 13S are required for early replication. The E1A 13S protein consists of 289 amino acids with four conserved regions (CR1–CR4) and can transactivate the other early transcription units E1B, E2, E3, and E4 [15], while E1A 12S lacks CR3.

E1 deleted adenoviruses are considered as replication-deficient and are used as shuttle vectors in gene therapy or for vaccination. Replication-competent adenoviruses that are restricted in their replication to cancer cells are therefore referred to as CRAd (Conditionally replicating Adenoviruses) [16] and are used clinically in oncolytic adenovirus therapy.

There are four general strategies in current clinical adenoviral vectors to achieve conditional replication: one concept is the replacement or overlay of the native E1 promoter with cancer cell-specific promoters, the other three concepts involve modification of the early transcription units E1A and E1B. In a number of constructs, an E1A modification is combined with a tumour specific promoter. However, there are several other promising approaches that have not yet been tested in clinical trials.

### 2.1. Specific Cellular Promoters Control E1A and Adenoviral Replication

Tumour-specific promoters are characterized by their ability to cause high expression of specific genes that are crucial for a malignant phenotype. Control of E1A-mediated viral replication by a tumour- or tissue-specific promoter restricts replication to the appropriate cells. Deletions in the E1A regulatory region should not interfere with either the inverted terminal repeat (ITR; 1–103 bp) or the packaging signal (194–358 bp) that overlaps with the E1A enhancer region.

Therefore, exogenous promoters are mostly inserted without any or with small deletions in the E1A regulatory region, leaving the E1A enhancer/encapsidation intact as in CRAd-Survivin-pk7 [17]. For further deletions, the packaging signal has to be rescued and cloned elsewhere as in the case of Ar6pAE2fE3F and CG0070. [18]. This allows tighter control of replication. The parallel regulation of the E4 region by a specific promoter also increases specificity, as the E4 gene products are necessary for productive viral replication [19]. To control E1A transcription, E2F-responsive elements [12], promoters for survivin [10], human telomerase reverse transcriptase (hTERT) [13] or chromogranin A (CgA) [20] were employed.

Restriction of adenoviral replication to specific tumour entities and tissues was also successfully achieved. For prostate tissue, the probasin promoter (to drive E1A) and the enhancer element of prostate-specific antigen (PSA) (to drive E1B) were used [21] and for hepatocellular carcinoma (HCC) the alpha-feto protein (AFP) promoter [22].

A synthetic fusion construct of core promoter and enhancers of the human tyrosinase was engineered for Ad2xTyr, an oncolytic adenovirus in which parts of the E1A regulatory region as well as the E4 promoter were replaced. Ad2xTyr showed tumour selectivity against melanoma cells versus normal fibroblasts and keratinocytes [19].

### 2.2. Modifications in E1A Control Specificity of Replication

Another approach to achieve tumour-specific adenoviral replication is to alter the protein-coding genes in the E1 region by introducing loss-of-function mutations. A cancer cell with specific molecular features can compensate for this defect, thus enabling replication.

#### 2.2.1. E1A Deletions: Delta 24

In a non-replicating cell, the retinoblastoma protein (pRB) interacts as a negative regulator with proteins of the E2F transcription factor family and thus control cell cycle progression [23]. This pathway is often genetically disrupted in tumour cells [24]. The CR2 region of the adenoviral E1A protein also interacts with pRB that results in release of E2F and enables viral replication [25,26]. In Delta-24, a deletion of 24 base pairs in the adenovirus E1A CR2 prevents the binding of E1A to pRB [27]. This virus cannot release E2F in normal cells and does not replicate. In tumour cells with mutated or dysregulated pRB, E2F is no longer negatively regulated by pRB and can activate viral gene transcription and thus replication [28].

In various further developments of Ad5-Delta-24, the modified E1A region is regulated via a tumour-specific promoter. In AdCN103, the 24 bp deletion of E1A is combined with a modified hTERT promoter [29] and in SynOV1.1 (Ad5-Delta-24-RGD-hGMCSF) with an enhanced cancer-specific alpha-fetoprotein (AFP) promoter. In other Ad5-Delta-24 derivatives, E1A transcription is controlled by one or more E2F-responsive elements such as in LOAd703 (Ad5/35-E2F-Delta-24) [30], ICOVIR-5 [31] and -7 (Ad5-E2F-Delta-24-RGD) [32], and VCN-01 (Ad E2F-Delta-24-RGD-PH20) [33].

#### 2.2.2. E1A 13S Deletion: Dl520, Ad-Delo3-RGD (XVir-N-31)

The adenoviral E2 promoter can be divided into an early and a late promoter for the regulation of the E2 transcription unit, which includes the DNA polymerase. The late E2 promoter region contains Y-boxes, binding sites for the transcription factor YB-1 (Y-box binding protein-1) [34,35]. A selective viral replication has been shown to be related to YB-1 expression [35]. Elevated YB-1 protein levels and subcellular localisation in the nucleus were demonstrated in malignant tissue and are associated with poor prognosis and multidrug resistance [36,37]. In normal adult cells, YB-1 is either not expressed or only localized in the cytoplasm [38,39]. Since E1A13S is critical for the translocalisation of YB-1 from the cytoplasm into the nucleus, adenoviruses lacking E1A13S expression are replication deficient in normal cells [40]. Dl520 is an oncolytic adenovirus in which E1A13S is not produced due to a deletion of 10 bp in the splice acceptor site of E1A CR3 [41]. A further development represents Ad-Delo3 RGD (XVir-N-31) that has additional deletions in E1B19K and E3 and an RGD motif in the fibre protein and is currently registered as a phase I clinical trial for glioblastoma [42,43] (EudraCT number: 2016-000292-25).

#### 2.2.3. E1B55K Deletion: Dl1520 (ONYX-015), Oncorine (H101)

The E1B region of the adenovirus genome encodes a 55-kD protein (E1B55K) that binds and inactivates p53. This binding is thought to be essential for virus replication, supposedly because E1A triggers p53-dependent apoptosis [44]. Dl1520 is a chimeric adenovirus (Ad5/2) with a deletion in E1B55K and a stop codon that terminates translation of the protein, and is identical to ONYX-015 [45,46]. This virus has limited tumour selectivity, thus challenging the concept [47,48]. However, based on this vector, the variant Oncorine (H101) was developed with an adenovirus 5 genome and partial deletion in E3 that has been approved for clinical application in China [49].

## 3. Oncolytic Adenoviruses and the Immune System

The induction of immunogenic cell death has proved to be a key aspect in virotherapy [50] (Figure 2). Based on this mechanism, oncolytic viruses can revert the tumour immune escape. It was shown that virus induced oncolysis results in apoptosis, necrosis, necroptosis, pyroptosis, or autophagy that releases signal molecules with a pathogen-associated molecular pattern (PAMP). Infected cells also release damage-associated molecules (DAMP), including HMGB1, heat shock proteins (HSP), intracellular matrix proteins, DNA, ATP, uric acid, or heparin sulphate [51,52]. These molecules stimulate the innate immune response and attract antigen-presenting cells (APCs). Tumour-associated antigens (TAA) and neoantigens in the tumour microenvironment can be taken up by APCs and trigger a systemic immune response against the tumour, so that a “cold” tumour can be transformed into a “hot” tumour that is recognised by the immune system [53]. The terms ‘oncolytic virotherapy’ and ‘oncolytic immunotherapy’ are therefore used synonymously. Oncolytic adenoviruses thus act multimodally against tumours by causing local inflammation, viral spread, destruction of the tumour microenvironment and release of tumour antigens, leading to a systemic anti-tumour response resulting in complete tumour eradication.

## 4. Arming of Adenoviral Vectors to Improve Their Therapeutic Effect

Besides tissue specific replication, vector systems have been developed that improve oncolytic and cytotoxic properties of adenoviruses. Thus, deletion of the E1B19K protein—a Bcl-2 homologue that blocks apoptosis induction—has been shown to increase viral spread and enhance TNF-alpha-mediated cell death [55,56].

### 4.1. Strategies to Include Transgenes in Adenoviral Vectors

Apart from strategies that warrant cancer-specific replication, genes that promote oncolysis or stimulate of the immune system have also been introduced.

Derived from early studies for the development of non-replicating vector systems, most oncolytic adenoviruses use deletions in the E3 region, as this region is non-essential for viral replication that allows inserts of up to 8.3 kb [10]. Expression cassettes are usually designed with exogenous promoters such as the human cytomegalovirus promoter or the heat shock protein 70 promoter [30,57]. In addition, ribosome entry sites, splice acceptor sites, or 2A self-cleaving peptide linkers can be used to regulate transcription of the transgene [58,59]. Another approach is to regulate inserted transgenes via the adenoviral E3 promoter and E3 polyadenylation site [60]. Here, the presence of the E3 adenoviral death protein (ADP) increases the oncolytic efficacy [61]. An alternative to the E3 region is the positioning of a transgene within the late genes which seems to have advantages in terms of specificity, expression level, and timing of expression. The insertion of hGM-CSF into the late gene L3 led to a high expression level of this transgene, which was moreover dependent on adenoviral DNA replication. In contrast, when the transgene was inserted into the E3 region, the expression was weak and independent of replication [62]. Similar observations were made when p53 was coupled with a late gene (L5, fibre) [63].

### 4.2. Transgenes That Confer Improved Therapeutic Effect

To enhance the therapeutic effect of oncolytic adenoviruses, various approaches have been employed such as gene-directed enzyme prodrug therapy (GDEPT), enhancement of virus-induced apoptosis and viral spread, expression of specific genes and attraction of the immune system.

The advantage of GDEPT is the local conversion of non-toxic compounds into cytotoxic drugs, thus minimizing systemic toxicity [64]. Some examples of enzymes used in oncolytic virotherapy are herpes simplex thymidine kinase in combination with ganciclovir or cytosine deaminase in combination with 5-fluorocytosine [65]. Other possibilities include the downregulation of cellular genes by RNA interference (RNAi) technologies [66]. The spread of oncolytic adenovirus particles within the tissue is mostly limited by the connective tissue. Therefore, viruses have been engineered to express enzymes that digest the extracellular matrix—e.g., human hyaluronidase to degrade hyaluronic acid or to express fusogenic membrane glycoproteins to induce cell fusion [67].

Bioselection of Ad5 mutants with enhanced anti-tumour activity in an in vivo tumour environment as selection pressure resulted in the variant AdT1 with a single adenine insertion within the retention domain of the endoplasmic reticulum (ER) of E3 gp19K that causes an enhanced release of the virus from the infected cell [68]. As described above, adenoviruses are generally immunogenic, but the effect can be enhanced by arming the oncolytic adenoviruses with transgenes encoding for immune stimulatory factors like TNF-α, IFNα, interleukin (IL)-2, IL-12, CD40L, the OX40 ligand (OX40L) [69] or the human granulocyte-macrophage colony-stimulating factor (GM-CSF) [70]. The 4-1BB ligand (4-1BBL, CD137L) was incorporated in LOAd703 [30] because it is known to enhance immunologic memory and expand natural killer (NK) cells [71]. For the treatment of HER2-positive cancer, the human trastuzumab heavy and light chain antibody encoding genes were inserted into an adenoviral vector and linked through an internal ribosome entry site. In the infected cell, the polypeptides were assembled into functional antibodies [72]. Immune checkpoint inhibitors (ICI), already successfully used in cancer therapy, have also been incorporated into adenoviral vectors in the form of transgenes encoding antibodies against the T lymphocyte-associated antigen-4 (CTLA-4) [73] or mini-antibodies against PD-L1 in a coinfected helper-dependent adenovirus (HDAd) [74].

## 5. Delivery and Tropism of Oncolytic Adenoviruses

Systemic application of adenoviral vectors leads to viral uptake in all organs and especially in the liver [75]. After intravenous administration, 90% of the virus is eliminated within the first 24 h by elements of the innate immune system [76]. Liver-resident macrophages (Kupffer cells) in particular efficiently take up and inactivate adenoviruses circulating in the blood [77], resulting in an estimated half-life in the blood of about two minutes [78]. A retention of the adenovirus in certain organs [79] can be achieved by direct injection into specific organs. However, within 14 days of Ad5 administration, a virus-specific humoral and cellular adaptive immune response occurs, eliminating virions and adenovirus-infected cells [80].

To ensure a potent anti-tumour response of oncolytic adenoviruses, modifications and delivery methods must be considered to achieve sufficient viral uptake into tumour cells.

### 5.1. Genetic Modifications to Alter the Tropism of Adenoviruses for Clinical Use

Cellular entry of adenoviruses is a two-step process consisting of binding of the viral fibre knob to the cell via the coxsackievirus and adenovirus receptor (CAR) [81] and interaction of Arg-Gly-Asp motifs (RGD) in the penton base with cellular integrins [82], leading to the internalisation of Ad5 vectors via clathrin-coated pits [83]. While integrins are homogeneously expressed in different tissues, CAR expression varies which largely affects the infectious capacity of andenoviruses [84]. For example, melanoma cells without CAR derived from metastases of patients were found to be resistant to WTAD infection and cell killing [85]. By integrating the RGD motif into the HI loop of the fibre knob domain [86] or into the hexon protein, enhanced infection of cells with low CAR expression was achieved [87]. A different strategy was used with the adenovirus CRAd-Survivin-pk7 in which a polylysine modification (Figure 3) with a heparan sulphate binding domain was incorporated into the fibre protein (pk7), resulting in high affinity for tumour-specific heparan sulphate proteoglycans [88].

### 5.2. Modification of Natural Tropism

Since adenoviral cell entry receptors are ubiquitously expressed in various tissues of mammals, one strategy to ensure tumour-exclusive therapy effects aims to ablate the natural viral tropism by altering the RGD motif in the penton base [89], incorporate mutations or deletions in the fibre protein, or by forming chimeras between two different adenovirus types [90]. A very commonly used adenovirus fibre chimera is a combination of the adenovirus serotype 3 fibre knob with an adenovirus serotype 5 shaft, thus recognizing CD46 instead of CAR [91,92].

To introduce a new tropism, Douglas et al. chemically conjugated an antibody fragment against the fibre protein with a folate molecule and obtained a targeted adenovirus [93]. The folate receptor is overexpressed on the surface of a variety of malignant cells. Similarly, tissue-specific targeting via the epidermal growth factor receptor (EGFR) [94], or via a tumour marker like melanoma-associated high molecular weight antigen [95], have been developed with the assistance of dimeric antibody fragments (diabodies) [96]. Another strategy includes the exchange of most of the fibre protein with an exogenous protein binding domain (affibody) [97] that has been specifically screened for high affinity binding to a tissue or tumour marker and for instance enables cellular attachment via HER2/neu rather than CAR [98].

### 5.3. Delivery Strategies

Besides local administration of oncolytic virus into the lesion, cell-mediated delivery is a novel and promising approach to improve the clinical usage of oncolytic adenoviruses. Here, neural [99] or mesenchymal [100] stem cells were infected with oncolytic adenoviruses in vitro and then delivered intratumourally or systemically into the patient. Tissue specific stem cells migrate to their appropriate tissue niche and would then distribute the virus, thus masking the viral vector to the host immune system and improve viral spread within the tumour [101].

## 6. Clinical Trials with Oncolytic Adenoviruses

Adenoviruses are the most commonly used oncolytic viruses in clinical trials [102]. At the time of writing this review, we conducted a search on clinicaltrials.org using the term “oncolytic adenovirus”, and subsequently also all individual names of the different adenoviral vectors mentioned, and identified 59 trials in various phases from 2005 to present. Of the 59 trials, 17 have been completed and 38 are ongoing. These trials are testing 18 different oncolytic adenoviruses administered intratumourally or intravenously. The majority were in phase I-II, three trials were in phase III and involved two oncolytic adenoviruses, Oncorine (H101) and CG0070. Table 1, Table 2, Table 3, Table 4 and Table 5 provides a detailed overview of the clinical trials collected on the website. We will only summarise the most advanced studies in this overview.

### 6.1. Clinical Vectors with Specific Cellular Promoters That Control E1A

Telomelysin (OBP-301) is an oncolytic adenovirus in which the hTERT promoter regulates both E1A and E1B, which are linked by an IRES element [103]. In an initial phase I dose escalation study in 16 patients with solid tumours, a single intratumoural injection resulted in a partial response in 56.7% of patients. Only grade 1/2 side effects were observed [104]. Phase II trials are currently ongoing for the treatment of patients with metastatic melanoma (NCT03190824) or adenocarcinoma of the oesophagus (NCT03921021). Among the first vectors using the tumour-selective E2F-1 promoter for E1A transcription are Ar6pAE2fE3F and Ar6pAE2fF (with and without the viral E3 region, respectively) [105]. These vectors were further enhanced, resulting in CG0070 (Ad5-E2F-E1A-E3-GMCSF) that expresses GMCSF in the E3 region and is currently tested in several clinical trials. In a 2018 phase II study published by [106], 45 patients with BCG-naïve non-muscle invasive bladder cancer were included. The overall response rate was 47% after 6 months and treatment was well tolerated. Interestingly, CIS may be the pathological subgroup that responds best to CG0070. The six-month complete response (CR) rates in patients with CIS and pure CIS were 50% and 58%, respectively. At six months, no patients with CIS-containing tumours developed muscle-invasive disease. The result is encouraging and promising, as progression occurred in approximately 9.8% to 40% of patients with intravesical BCG treatment. A phase III trial of CG0070 as monotherapy started in 2020 (NCT02365818), and two phase I-II trials in combination with chemotherapy are ongoing.

### 6.2. Clinical Vectors with E1A Deletion: Ad5-Delta-24

The first AdDelta-24 derivative being tested in clinical trials was DNX-2401 (Delta-24-RGD) [16] mainly for the treatment of glioma and has already completed four phase I trials (NCT01582516, NCT01956734, NCT02798406, NCT02197169) with reported improved long-term survival and evidence of effective immune activation. Five of 25 patients survived more than three years after treatment, and three patients experienced a dramatic decrease in tumour size, resulting in a progression-free survival of three years [107]. This was followed by six phase I/II studies and confirmed the safety profile. In further studies, the original oncolytic construct was combined with other drugs or/and further developments of AdDelta-24 were used. A phase II trial of DNX-2401 in combination with pembrolizumab, an anti-PD-1 antibody, included 49 patients with recurrent glioblastoma (NCT02798406). Median overall survival was 12.5 months and survival at 18 months was 20.2% compared to the median overall survival of 7.2 months with monotherapies of iomustine and temozolomide, the results with the oncolytic adenovirus were encouraging [108].

Table 1, Table 2, Table 3, Table 4 and Table 5 Clinical trials with oncolytic adenoviruses up to the time of writing, according to searches on clinicaltrials.org from 2005 to August 2021. 59 trials were found in various phases from 2005 to date. Of these 59 trials, 17 have been completed and 38 are ongoing. These trials are testing 18 different oncolytic adenoviruses. In Table 1, Table 2, Table 3 and Table 4, the clinical trials are grouped according to the concepts for conditionally replicating adenoviruses explained in Figure 1, while Table 5 lists the clinical trials with enadenotuvirev (Colo-Ad1).

### 6.3. Clinical Vectors with Specific Cellular Promoters That Control E1A-Delta-24

ICOVIR-5 (Ad5-E2F-Delta-24-RGD) was administered systemically intravenously in a phase I trial in melanoma patients (NCT01864759) and was well tolerated [109]. Although there was no tumour regression in a total of 12 patients, viral DNA was detected in metastatic skin or liver lesions in four patients, suggesting that ICOVIR-5 can target and detect metastatic tumour cells when administered intravenously.

VCN-01 (Ad5-E2F-Delta-24-RGD-PH20) expresses hyaluronidase (PH20), which enhances intratumoural spread of the virus [33]. It is currently being tested in several clinical trials in combination with chemotherapy or immune checkpoint inhibitor in advanced pancreatic cancer (NCT02045589 and NCT02045602) and squamous cell carcinoma of the head and neck (NCT03799744). In a trial for retinoblastoma (NCT03284268), administration of VCN-01 was well tolerated and exhibited anti-tumour activity in retinoblastoma vitreous seeds [110].

### 6.4. Clinical Vectors with E1B55K Deletions

ONYX-015 (dl1520) was the first oncolytic adenovirus in clinical trials for the treatment of head and neck cancer [111]. Although no objective response was observed, tumour necrosis at the injection site was detected in 5 of 22 patients. Phase I clinical trials for Oncorine (H101) were initiated in China in 2000. In addition to a tolerable safety profile, 3 of 15 patients reported remarkable tumour shrinkage. The phase III trial showed that Oncorine achieved a better positive response rate (79%) in patients with squamous cell carcinoma of the head and neck (SCCHN) in combination with chemotherapy than with chemotherapy alone (40%) [112]. Based on this study, it was approved by the State Food and Drug Administration (SDFA) of China in 2005 as the world’s first commercialised oncolytic virus for the treatment of SCCHN in combination with chemotherapy.

### 6.5. Clinical Vectors Based on Direct Evolution

A novel adenoviral vector, enadenotucirev (Colo-Ad1), is derived from a pool of different serotypes of species B to F by directed evolution and was selected to replicate only in colon cancer cells [113]. Three phase I trials were conducted. One of them included 17 patients with solid tumours and examined the efficacy of intratumoural versus intravenous application. Viral DNA was detected in tumour samples in 11 of 12 patients after intravenous infusion and in 2 of 5 after intratumoural injection. Both methods were well tolerated and no treatment-related serious adverse events were reported. Two other phase I trials with enadenotucirev are currently recruiting patients with colon cancer, head and neck cancer, or other epithelial tumours for combination therapy with enadenotucirev and nivolumab (PD-1 inhibitor) (NCT02636036) or patients with rectal cancer (NCT03916510) to be treated in combination with radiotherapy and chemotherapy (capecitabine). To elicit further immune responses against the tumour, two variants of enadenotucirev are currently being investigated in phase I clinical trials: NG-350A (NCT03852511), which expresses the CD40 antibody and NG-641 (NCT04053283), which expresses the bispecific T-cell engager (BiTE) FAP/CD3 chemokine ligands 9 and 10 (CXCL9 and CXCL10) and interferon alpha (IFNα).

## 7. Conclusions and Future Perspectives

The clinical application of therapies based on oncolytic adenoviruses has proven their enormous potential, but is still in its infancy. In particular, the wide-ranging compatibility with already established therapies (without increased toxicity), the multiple cell-killing effects, and the induction of an immunogenic cell death makes it highly attractive in the clinical arena.

Three aspects are of crucial relevance in the generation of novel viral vectors: (a) tissue specificity, (b) replication and lysis capability; and (c) triggering of the immune system in order to achieve a systemic therapy response. There are several sophisticated methods to increase the selectivity of infection of tumour cells to obtain a higher level of safety and efficacy—e.g., by adding certain transgenes. However, selectively replicating oncolytic adenoviruses are often attenuated in their replicative capacity compared to wild type [114]. This not only affects lysis of the tumour but also activation of the immune system. Combination with small molecules, interacting with signalling pathway, such as the JAK/STAT pathway [115], radiation [116], or chemotherapy, are possible strategies to address this issue [117].

The most promising aspect of the development of new adenoviral vectors is the activation of the immune system by the virus, which would represent a vaccination strategy against the tumour. A better understanding of mutually supporting effects of introducing certain transgenes or combining them with immune checkpoint inhibitors will allow novel directions in treatment strategies to increase oncolytic or immune stimulatory efficacy. Several clinical trials are currently underway, and it will gradually become clear which strategies and modifications will provide the best results. An important aspect in these trials will be a thorough patient stratification that addresses the characteristics of the used oncolytic adenovirus. As we have only recently begun to translate the knowledge gained mainly in the laboratory into the clinic, new findings will automatically generate a great deal of interest in the enormous possibilities offered by the great versatility in developing new treatment strategies for cancer with adenoviruses.

## Figures and Tables

**Figure 1 ijms-22-10522-f001:**
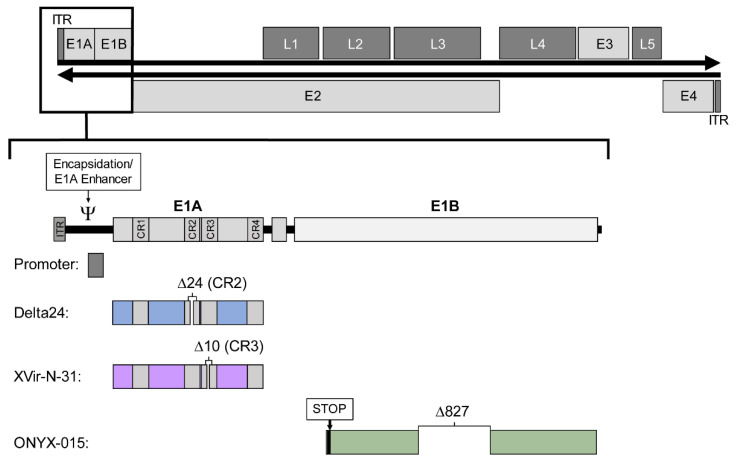
Structure of the wild-type adenovirus serotype 5 genome and the four concepts for generating conditionally replicating adenoviruses applied in vectors of current clinical trials. The affected E1 region is shown enlarged to illustrate the changes. The double-stranded DNA of about 36 kb in length (black horizontal arrows), flanked by two inverted terminal repeats (ITR) comprises four early transcription units (E1–E4, light grey boxes) and five late transcription units (L1–L5, dark grey boxes). The E1 region (enlarged below) consists of the encapsidation signal/E1A enhancer, the E1A promoter, E1A with four conserved regions (CR1–CR4) and E1B. The four concepts are listed below: Changes in the promoter (grey), Delta24 (blue) with a deletion in CR2, XVir-N-31 (purple) with a deletion in CR3, and ONYX-015 (green) with a stop codon and a deletion in E1B55K.

**Figure 2 ijms-22-10522-f002:**
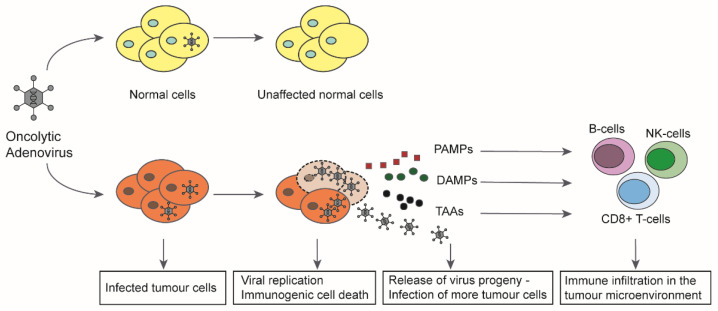
Mechanism of action of oncolytic adenoviruses in cancer cells. In normal, non-cancerous cells, oncolytic adenoviruses are unable to replicate upon infection leaving the cells unaffected. In tumour cells, the viruses can successfully replicate upon infection, leading to production of more virus progeny and eventual induction of immunogenic cell death (ICD). The cell lysis leads to release of virus progeny, pathogen-associated molecular patterns (PAMP), damage-associated molecular patterns (DAMP), and tumour associated antigens (TAA) into the tumour microenvironment (TME). The released virus progeny further infects the uninfected tumour cells and continues the viral spread. The immunostimulatory molecules released into the TME attract immune cells to the tumour and lead to the activation of both innate and adaptive immunity against the tumour (modified with permission from [52,54]).

**Figure 3 ijms-22-10522-f003:**
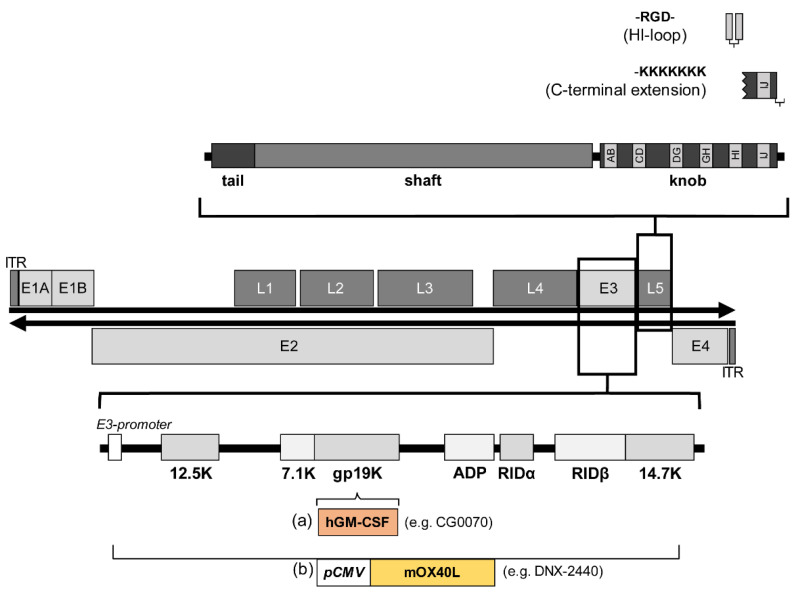
Structure of the wild-type adenovirus serotype 5 genome and the concepts for altering natural tropism by fibre modification (upper section) and incorporating transgenes into the adenoviral E3 region (lower section). The affected L5 region encodes for the fibre protein; fibre monomer with tail, shaft, and knob domains is enlarged in the upper section to illustrate the modifications. Incorporation of an Arg-Gly-Asp motif (RGD) into the HI-loop enables attachment to the host cell via integrins, and C-terminal extension of lysin residues at the fibre head via heparansulfate-containing receptors. The lower section illustrates the two concepts of incorporating transgenes into the E3 region: (a) Most of the E3 region is retained while a transgene like the human granulocyte-macrophage colony-stimulating factor (GM-CSF) in the adenoviral vector CG0070 is inserted, here GM-CSF expression is regulated by the E3 promoter. (b) Replacement of the E3 transcription unit with, for example, the pCMV-mOX40L-expression cassette in the adenoviral vector DNX-2440 where the expression of the mouse OX40L is regulated by the cytomegalovirus (CMV) promoter.

**Table 1 ijms-22-10522-t001:** Clinical trials using oncolytic adenoviruses whose replication is controlled by cellular promoters.

Concept: Promoter	Indication	Clinical Trials.gov No.	Vector Name	Construct	Additional Features	Pre-Clinical	I	II		III	Combination	Application of AdV
E2F1- promoter	Bladder Cancer	NCT00109655	CG0070	Ad5-E2F-E1A-GMCSF	GM-CSF	2005						Intravesical
		NCT02365818				2015 completed						Intravesical
		NCT04452591				2020 recruiting						Intravesical
		NCT04610671				2020 recruiting					Nivolumab	Intravesical
		NCT04387461				2020 recruiting					Pembrolizumab	Intravesical
hTERT- promoter	Hepatocellular Carcinoma	NCT02293850	OBP-301	Ad5-hTERT-E1A-IRES-E1B		2014 unknown						Intratumoural
	Melanoma	NCT03190824				2017 active						Intratumoural
	Solid Tumours	NCT03172819				2017 active					Pembrolizumab	Intratumoural
	Esophageal Cancer	NCT03213054				2017 recruiting					Radiation	Intratumoural
	Esophagogastric Adenocarcinoma	NCT03921021				2019 recruiting					Pembrolizumab	Intratumoural
	Head and Neck Squamous Cell Carcinoma	NCT04685499				2020 recruiting					Pembrolizumab	Intratumoural
	Esophageal and Gastroesophageal Junction Adenocarcinoma	NCT04391049				2020 not recruiting					Paclitaxel, Carboplatin, Radiation	Intratumoural
Survivin- promoter	Glioma	NCT03072134	CRAd-S-pk7	NSC-CRAd-Survivin-pk7	pk7	2017 active						Intratumoural
CgA- promoter	Neuroendocrine Tumours	NCT02749331	AdVince	CgA -PTD-miR122	PTD-peptide, miRNA 122	2016 recruiting						Intraarterial

**Table 2 ijms-22-10522-t002:** Clinical trials using oncolytic adenoviruses in which a delta 24 variant of E1A is controlled by cellular promoters.

Concept: Promoter & Δ24	Indication	Clinical Trials.gov No.	Vector Name	Construct	Additional Features	Preclinical	I	II		III	Combination	Application of AdV
AFP promoter	Hepatocellular Carcinoma	NCT04612504	SynOV1.1	Ad5-AFP-delta24-RGD	GM-CSF	2020 recruiting					Atezolizumab	Intratumoural
E2F1 promoter	Melanoma	NCT01864759	ICOVIR-5	Ad5-E2F-delta24-RGD		2013 completed						Intravenous
	Solid Tumours	NCT01844661			Mesenchymal allogenic cells	2013 completed						Intravenous
	Glioma	NCT04758533				2021 not recruiting						Intravenous
	Advanced Solid Tumours Pancreatic Adenocarcinoma	NCT02045602	VCN-01	Ad5-E2F-delta24-RGD -PH20	Hyaluronidase (PH20)	2014 completed					Gemcitabine, Abraxane	Intravenous
	Pancreatic Adenocarcinoma	NCT02045589				2014 completed					Gemcitabine, Abraxane	Intravenous
	Refractory Retinoblastoma	NCT03284268				2017 recruiting						Intravitreal
	Squamous cell Carcinoma of Head and Neck	NCT03799744				2019 recruiting					Durvalumab	Intravenous
	Pancreatic Cancer	NCT02705196	LOAd-703	Ad5/35-E2F-delta24	CD40L, 4-1BBL	2016 recruiting					Gemcitabine, Nab-Paclitaxel, Atezolizumab	Intratumoural
	Pancreatic Cancer/Ovarian, biliary, colorectal Cancer	NCT03225989				2017 completed					Standard chemotherapy or Gemcitabine	Intratumoural
	Colorectal Cancer	NCT03555149				2018 recruiting					Standard chemotherapy	Intratumoural
	Melanoma	NCT04123470				2019 recruiting					Atezolizumab	Intratumoural

**Table 3 ijms-22-10522-t003:** Clinical trials using oncolytic adenoviruses with a 24 base pair deletion in E1A.

Concept:	Indication	Clinical Trials.gov No	Vector Name	Construct	Additional Features	Preclinical	I	II		III	Combination	Application of AdV
Δ24	Ovarian Cancer, Primary Peritoneal Cancer	NCT00562003	Ad5-Delta 24RGD			2017 completed						Intraperitoneal
	Brain Cancer	NCT00805376	DNX-2401	Ad5-Delta 24RGD		2018 completed						Intratumoural
	Glioblastoma	NCT01582516				2012 completed						Intratumoural
	Glioblastoma or Gliosarcoma	NCT01956734				2013 completed					Temozolomide	Intratumoural
	Brain Cancer	NCT02798406				2016 completed					Pembrolizumab	Intratumoural
	Glioblastoma or Gliosarcoma	NCT02197169				2014 completed					Interferon-gamma	Intratumoural
	Gliomas	NCT03178032				2017 active, not recruiting					Radiotherapy, chemotherapy	Intratumoural
	Gliomas	NCT03896568	Ad5-Delta 24RGD		Mesenchymal stem cells as carriers	2019 recruiting						Intraarterial
	Gliomas	NCT03714334	DNX-2440	Ad5-delta24-RGD-OX40L	CMV-OX40L-BGHpA repl. E3	2018 recruiting						Intratumoural
	Glioblastoma and multiple Solid Tumours	NCT04714983				2021 recruiting						Intratumoural
	Solid Tumours	NCT01598129	ONCOS-102	Ad5/3-delta24-GMCSF	GM-CSF	2012 completed					Cyclophosphamide	Intratumoural and intravenous
	Melanoma	NCT03003676				2016 active, not recruiting					Cyclophosphamide, Pembrolizumab	Intratumoural
	Colorectal, Ovarian, Appendiceal Cancer	NCT02963831				2017 active, not recruiting					Durvalumab	Intraperitoneal
	Colorectal, Ovarian, Appendiceal Cancer	NCT03514836				2018 active, not recruiting					Durvalumab	Intratumoural
	Prostate Cancer	NCT04097002	ORCA-010	Ad5-delta24-RGD	E3/19K-T1 protein	2019 recruiting						Intratumoural
	Melanoma	NCT04217473	TILT 123	Ad5/3-delta24-TNFα-IRES-IL2	TNFα-IRES-IL2 in E3	2020 recruiting						Intratumoural
	Melanoma, Solid Tumours	NCT04695327				2021 recruiting						Intratumoural
	Diverse HER2 positive Solid Tumours	NCT03740256	CAdVec	Ad5-Delta 24	HER2-specific autol. CAR T cells	2018 recruiting					Helper dependent Ad expressing PD-1 minibody	Intratumoural

**Table 4 ijms-22-10522-t004:** Clinical trials using oncolytic adenoviruses with E1A13S or E1B55K deletions.

Concept:	Indication	Clinical Trials.gov No.	Vector Name	Construct	Additional Features	Preclinical	I	II	III	Combination	Application of AdV
ΔE1A 13S	Glioblastoma	2016-000292-25 (EudraCT no.)	XVir-N-31	Ad5 E1A13S/E1B19K/E3 deletions, Fibre RGD		2021 active, not recruiting					Intratumoural
ΔE1B 55K	Nasopharyngeal Carcinoma	no number		Ad5 E1B55K/E3 deletions		2005 approved by the State Food and Drug Administration of China				Cisplatin, 5-Fluorouracil, Adriamycin	Intratumoural
	Refractory Malignant Ascites	NCT04771676	Oncorine (H101)			2021 recruiting					Intraperitoneal
	Lung Cancer	NCT02579564				2015 active, not recruiting				Gemcitabine, Vinorelbine, Paclitaxel, Pemetrexed, Endostar, Cisplatin	Intrathoracic
	Hepatocellular Carcinoma	NCT03780049				2018 recruiting				Oxaliplatin, 5-Fluorouracil, Leucovorin	Intraarterial
	Prostate Cancer	NCT02555397	Ad5-yCD/ mutTKSR39rep-hIL12	E1B55K deletion	yCD, mutTk, IL-12 *	2015 unknown					Intraprostatic
	Metastatic Pancreatic Cancer	NCT03281382				2017 unknown				5-Fluorocytosine (5-FC), chemotherapy	Intratumoural

* yCD = yeast cytosine deaminase, HSV-1 TKSR39/mutTk = mutant form of herpes simplex virus type 1 thymidine kinase.

**Table 5 ijms-22-10522-t005:** Clinical trials using the vector enadenotucirev (Colo-Ad1), a vector whose specificity is based on selection in colon carcinoma.

Other Approaches	Indication	Clinical Trials.gov No.	Vector Name	Construct	Additional Features	Preclinical	I	II		III	Combination	Application of AdV
Evolution	Solid Tumours	NCT02053220	Enadenotucirev (Colo-Ad1)	Ad3/11 Chimera		2014 completed						Intravenous
	Solid Tumours of Epithelial Origin	NCT02028442				2014 completed						Intravenous
	Ovarian Cancer	NCT02028117				2014 completed						Intravenous
	Colorectal Cancer, Squamous Cell Carcinoma of the Head and Neck, Epithelial Tumours	NCT02636036				2015, active not recruiting					Nivolumab	Intravenous
	Rectal Cancer	NCT03916510				2019 recruiting					Capecitabine, radiation	Intravenous
	Epithelial Tumours	NCT03852511	NG-350A		Anti-CD40 Ab	2019 recruiting						Intratumoural, intravenous
	Epithelial Tumours	NCT04053283	NG-641		FAP/CD3, CXCL9, CXCL10, IFNa	2019 recruiting					chemotherapy, checkpoint inhibitors	Intratumoural, intravenous
	Squamous Cell Carcinoma of the Head and Neck	NCT04830592	NG-641			2021 active, not recruiting					Pembrolizumab	Intravenous

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
