# Peer review of "Concepts in Oncolytic Adenovirus Therapy"

_ijms, 2021, doi:10.3390/ijms221910522_

Round 1

Reviewer 1 Report

The review is well written and very exhaustive. The authors probably could shorten the clinical trials by including the most promising ones.

Reviewer 2 Report

This is a narrative review focused on genetic changes important for tumour-specific replication, transgene expression, interaction with the immune system and summarizes the ongoing clinical trials.

It has strong points, but authors should made the following amendments:

Although it is a narrative review, the abstract section should obtain a more structured format.

The introduction section should contain the current gap in the literature leading to the rationale of the current paper.

There must be a search strategy and a schedule from the start till the end of the paper, together with a separate section with the limitations of the study.

Authors should also indicate the originality of their figures.

The conclusion should adhere to their exact findings.

Round 2

Reviewer 2 Report

The 3 first comments should be revisited.

As for the modification of Cunliffe et al, 2020, authors should state that they have had the relevant permission.
